# Respiratory Outcomes of Insulin Use in Patients with COPD: A Nationwide Population-Based Cohort Study

**DOI:** 10.3390/ph16050643

**Published:** 2023-04-24

**Authors:** Fu-Shun Yen, Shu-Hao Chang, James Cheng-Chung Wei, Ying-Hsiu Shih, Chii-Min Hwu

**Affiliations:** 1Dr. Yen’s Clinic, No. 15, Shanying Road, Gueishan District, Taoyuan 33354, Taiwan; yenfushun@gmail.com; 2Division of Pulmonary Medicine, Department of Internal Medicine, Cheng Ching Hospital, No. 139, Pingdeng Street, Central District, Taichung 40045, Taiwan; ccgh7458@yahoo.com.tw; 3Department of Allergy, Immunology & Rheumatology, Chung Shan Medical University Hospital, No. 110, Section 1, Jianguo N. Road, South District, Taichung 40201, Taiwan; jccwei@gmail.com; 4Institute of Medicine, Chung Shan Medical University, Taichung 40201, Taiwan; 5Graduate Institute of Integrated Medicine, China Medical University, Taichung 40402, Taiwan; 6Management Office for Health Data, China Medical University Hospital, Taichung 40201, Taiwan; hsiu.cmuh@gmail.com; 7College of Medicine, China Medical University, Taichung 40402, Taiwan; 8Faculty of Medicine, National Yang-Ming Chiao Tung University School of Medicine, No. 155, Section 2, Linong Street, Taipei 11221, Taiwan; 9Section of Endocrinology and Metabolism, Department of Medicine, Taipei Veterans General Hospital, No. 201, Section 2, Shipai Road, Beitou District, Taipei 11217, Taiwan

**Keywords:** mortality, bacterial pneumonia, non-invasive positive pressure ventilation, invasive mechanical ventilation, hypoglycemia

## Abstract

Acute exacerbations of chronic obstructive pulmonary disease (COPD) with severe hyperglycemia may require insulin to lower glucose levels in people with coexisting type 2 diabetes (T2D) and COPD. We conducted this study to examine the risk of hospitalization for COPD, pneumonia, ventilator use, lung cancer, hypoglycemia, and mortality with and without insulin use in people with T2D and COPD. We adopted propensity-score-matching to identify 2370 paired insulin users and non-users from Taiwan’s National Health Insurance Research Database between 1 January 2000 and 31 December 2018. Cox proportional hazards models and the Kaplan–Meier method were utilized to compare the risk of outcomes between study and control groups. The mean follow-up for insulin users and non-users was 6.65 and 6.37 years. Compared with no insulin use, insulin use was associated with a significantly increased risk of hospitalization for COPD (aHR 1.7), bacterial pneumonia (aHR 2.42), non-invasive positive pressure ventilation (aHR 5.05), invasive mechanical ventilation (aHR 2.72), and severe hypoglycemia (aHR 4.71), but with no significant difference in the risk of death. This nationwide cohort study showed that patients with T2D and COPD requiring insulin therapy may have an increased risk of acute COPD exacerbations, pneumonia, ventilator use, and severe hypoglycemia without a significant increase in the risk of death.

## 1. Introduction

Genetic factors, early-life events, infections, smoking, and environmental pollutants are the risk factors for developing chronic obstructive pulmonary disease (COPD) [1,2,3]. These factors can lead to lung damage or remodeling of the airways, pulmonary parenchyma, and vasculature, resulting in progressive airway limitation, dyspnea, exercise intolerance, restricted daily activity, and premature mortality [1,2]. Unstable COPD or infections can lead to episodic acute exacerbations. Each exacerbation contributes to pulmonary function deterioration and patient suffering [1,4,5]. The number of people with COPD worldwide has increased from 142 million to 212 million (33%) between 1999 and 2019 [6]. COPD is also the third leading cause of death worldwide [1,2]. COPD is usually associated with cardiovascular diseases and type 2 diabetes (T2D), possibly due to shared risk factors (aging, physical inactivity, and smoking) [1,2,5,7]. The frequent use of corticosteroids and low-grade systemic inflammation in people with COPD also predispose them to T2D. People with COPD have elevated levels of plasma CRP and nuclear factor-κB (NF-κB) activation, which can attenuate the suppression of hepatic glucose production by insulin and can induce insulin resistance in peripheral tissues. Pulmonary inflammation can spill over into systemic inflammation and adversely influence systemic glucose homeostasis by reducing the recruitment of skeletal muscle capillaries that deliver glucose and insulin to muscle cells, to increase blood glucose levels [1,7,8,9]. Approximately 16% of patients with COPD have T2D, and T2D can worsen the prognosis of COPD [1,7,8,10]. However, there is little research and progress on treating comorbid T2D and COPD [1,2,7].

Insulin was discovered in 1921 and has been in use to treat and save many people with type 1 diabetes (T1D), with little or no insulin production [11]. It also is the most powerful glucose-lowering medicine in adult persons with T2D [12]. Insulin is indicated for patients with severe hyperglycemia (random glucose >300 mg/dL), symptomatic diabetes, ketoacidosis, acute medical events, concomitant diseases, admission, and inability to take oral antidiabetic drugs [12]. Randomized prospective studies have shown that insulin effectively lowers blood glucose and reduces microvascular complications [13,14]. One report indicates that insulin has potential anti-inflammatory, anti-oxidant, and anti-apoptotic effects [15]. Patients with cystic fibrosis have less insulin secretion and are more likely to have diabetes and require insulin therapy. They often have poorer nutrition, greater catabolism and a higher risk of infection. This can have a negative impact on the function of their lungs and their chances of survival [16]. COPD often coexists with other comorbidities and can lead to acute exacerbations requiring urgent care [4,5]. Currently, there are no guidelines for managing diabetes mellitus in persons with T2D and COPD. The recommendation for antidiabetic treatment in these patients is in accordance with the guidelines for the management of diabetes mellitus [7,17]. Patients with acute exacerbations of COPD can be treated with oral antidiabetic agents or glucagon-like peptide-1 agonist if their blood glucose levels are not very high. However, if they have severe hyperglycemia, acute symptoms or require hospitalization, they may need insulin or follow inpatient diabetes management guidelines [18]. They can, therefore, be treated initially with long-acting insulin analogues because they are more stable, have a longer lasting effect and have a lower risk of glucose fluctuations and hypoglycemia than neutral protamine Hagedorn insulin and premixed insulin [17]. Rapid-acting insulin analogues can then be added as needed, as they can rapidly lower blood glucose levels with less risk of postprandial hypoglycemia than regular short-acting insulin [17]. Combined with tight titration of insulin doses and blood glucose monitoring, insulin treatment can be a safe and effective antidiabetic management strategy in persons with COPD and high blood glucose levels [7,18]. Both T2D and COPD are metabolic disorders. Insulin is a drug candidate for treating severe hyperglycemia in people with T2D and COPD. However, no research has studied the long-term results of using insulin in these patients [7,8,9]. Therefore, we conducted this cohort study to compare the outcomes of all-cause mortality, hospitalization for COPD, bacterial pneumonia, ventilator use, lung cancer, and severe hypoglycemia in people with coexisting T2D and COPD with or without receiving insulin therapy.

## 2. Results

### 2.1. Participants

From the 1 January 2000 until the 31 December 2018, we identified 28,399 insulin users and 25,592 nonusers among persons with coexisting T2D and COPD. After excluding unsuitable patients, there were 3490 insulin users and 5667 nonusers. Finally, we identified 2370 insulin users and 2370 nonusers through 1:1 matching of the propensity scores (Figure 1).

The basic characteristics, comorbidities, and medication were well matched between the study group and the control group with SMD values <0.1 (Table 1). In the comparison cohorts of the insulin users and nonusers, the male proportion was 62.45% and 62.28%, respectively; the mean (SD) age was 61.02 (11.11) and 60.73 (11.21) years, respectively; the mean follow-up time was 6.65 (4.34) and 6.37 (4.29) years in each case.

### 2.2. Main Outcomes

In the cohorts that were matched, 357 (15.06%) insulin users and 326 (13.76%) insulin non-users died during the follow-up period (incidence rate: 22.28 versus 21.45 per 1000 person-years). Multivariable analysis showed that the adjusted HR (aHR) for insulin users vs. non-users was 1.08 (95% CI = 0.93–1.26, *p* = 0.3076; Table 2). However, compared with nonusers, insulin users had a significantly higher risk of hospitalization for COPD (aHR 1.7, 95% CI 1.24–2.32), bacterial pneumonia (aHR 2.42, 95% CI 1.95–3), NIPPV (aHR 5.05, 95% CI 2.76–9.22), IMV (aHR 2.72, 95% CI 1.99–3.72), and severe hypoglycemia (aHR 4.71, 95% CI 2.5–8.89). 

The cumulative incidence of hospitalization for COPD (Log-rank test *p*-value < 0.001), bacterial pneumonia (Log-rank test *p*-value < 0.001), and IMV (Log-rank test *p*-value < 0.001) were significantly higher in insulin users than in non-users (Figure 2).

### 2.3. Dose-Response Analysis

The cumulative duration (<90, 90–179, >179 days) of insulin use had significantly increased risks of hospitalization for COPD, bacterial pneumonia, and IMV than insulin no-use, and longer cumulative insulin use appeared to increase the risk of these outcomes (Table 3).

## 3. Discussion

This nationwide cohort study demonstrated that patients with coexisting T2D and COPD requiring insulin therapy might increase the risk of hospitalization for COPD, bacterial pneumonia, ventilation use, and severe hypoglycemia; however, it did not increase the risk of death. Longer cumulative insulin use had association with increased risk of hospitalization for COPD, bacterial pneumonia, and invasive mechanical ventilation than insulin no-use.

Acute attacks of COPD often require hospitalization and can lead to respiratory failure or death [1,2,19]. The goal of COPD treatment is to reduce acute exacerbations of COPD [1,4,5]. One report showed that insulin sensitizers such as metformin may reduce the risk of severe COPD exacerbations [20]. Hyperglycemia tends to impair pulmonary function and increase the risk of acute COPD attacks [7,8,10]. Although insulin is the most effective glucose-lowering drug, our study showed that it might increase the risk of hospitalization for COPD. Insulin treatment has a direct effect on the lungs by inducing prostaglandin-mediated contraction of airway smooth muscle [21]. Insulin-like growth factor-1 (IGF-1) appears to play an important role in lung disease, but circulating IGF-1 has been reported to be associated with a lower risk of airway hyperresponsiveness [22]. Using insulin is likely to cause hypoglycemia and weight gain [8,23]. Prolonged use of insulin can lead to hyperinsulinemia, downregulation of insulin signaling and insulin resistance, which can increase glucose levels and oxidative stress. The downregulated insulin signaling leads to an imbalance in the anabolic activity of insulin, favoring protein synthesis and suppressing autophagy. The latter promotes cell senescence by inhibiting autophagic protein and lipid degradation and turnover [24]. All these factors can worsen lung function and exacerbate COPD.

Patients with COPD may exhibit alteration of the airway microbiome and impaired airway immune response, thus increasing susceptibility to bacterial infections [1,3,19]. Hyperglycemia tends to increase the permeability and spread of bacteria through the airways [8,25]. Diabetes mellitus is associated with a higher risk of bacterial pneumonia due to impaired cellular immunity [7,10]. We have previously shown that thiazolidinediones may increase the risk of developing bacterial pneumonia in people with COPD [26]. This study also found that insulin may increase the risk of bacterial pneumonia in people with T2D and COPD. The reason for this result is unclear. However, previous research has shown that hyperinsulinemia or insulin resistance can increase proinflammatory cytokines, predispose patients to inflammation, and impair immune function, thus increasing infection risk [23].

Hypoxia or respiratory failure can be a medical emergency in patients receiving COPD treatment [1] and often requires noninvasive ventilation to improve the patient’s condition [2,4,19]. Intubation with invasive mechanical ventilation can be a life-saving procedure in patients with persistent hypercapnia [4]. This study showed that the use of insulin was associated with a higher risk of both NIPPV and IMV in people with COPD, and longer cumulative insulin use was associated with further increased risk of these outcomes. The use of insulin in patients with coexisting COPD and T2D may require attention to the possibility of respiratory failure.

Smoking is the main cause of lung cancer [1,27,28,29]. Both COPD and T2D are risk factors for lung cancer [1,8,19]. A meta-analysis has shown that metformin may reduce the risk of lung cancer development [30]. Insulin can promote cell proliferation and mediate a mitogenic effect as binding to the insulin receptor. Insulin receptors can also bind to the insulin-like growth factor with potential oncogenic effects [31]. However, this finding has not been proved in large, long-term randomized controlled insulin trials [32]. Our study showed that insulin was associated with a non-significantly increased risk of lung cancer (aHR 2.11, 95%CI 0.99–4.49) in people with T2D and COPD. Insulin therapy in patients with COPD may need close monitoring for the occurrence of lung cancers.

Almost all studies on insulin therapy demonstrate the side effect of hypoglycemia [12,13,14]. Previous study has shown that about 37% of patients receiving insulin therapy developed hypoglycemia, and about 2.3% of patients with T2D had severe hypoglycemia annually [14]. Hypoglycemia can cause symptoms of cold sweating, palpitation, dyspnea, and general weakness. Kasirye et al. showed that hypoglycemia resulted in more severe acute exacerbation in patients with COPD [33]. Our research revealed that insulin use in patients with coexisting T2D and COPD could increase the risk of severe hypoglycemia. Dyspnea, weakness, acute exacerbation of COPD, and ventilation use could be attributed to severe hypoglycemia in patients on insulin therapy in this study. Elliot P Joslin stated, “Insulin is a remedy primarily for the wise and not for the foolish, be they patients or doctors… to use insulin successfully requires more brains” in 1928 [34]. Insulin therapy requires effective doctor-patient communication and careful monitoring for hypoglycemia.

COPD has been the third leading cause of death worldwide for years [1,4]. Diabetes mellitus can increase the severity of COPD and the risk of death [1,7,10]. Therefore, people with coexisting T2D and COPD are at high risk of death. Previous study has shown that metformin may have an effect on the risk of death in these patients [35]. Empagliflozin has been shown to have consistent benefits on cardiopulmonary and mortality outcomes in patients with T2D and COPD [36]. Our study showed that although insulin use might increase the risk of severe hypoglycemia, bacterial pneumonia, and ventilation use, it might not increase the risk of death in patients with T2D and COPD. Insulin studies tend to recruit patients with a longer duration of diabetes or more severe cases with a higher mortality rate. However, it is possible that this condition is not prominent in this study.

This study had several advantages. First, this was a nationwide population-based study. As 99% of Taiwan’s 23 million people are covered by National Health Insurance, the patients enrolled in the study had less selection bias. Second, the study had a large sample size and an observational period of 18 years, which allowed for subgroup analysis and assessment of outcomes that took years to develop. Third, although this was only an observational study, no known randomized insulin studies have been conducted in patients with COPD recently, and the possibility of other randomized studies on insulin use in patients with COPD is low in the future [1,7,8,9]. Therefore, this real-world data may fill some knowledge gaps in the clinical care of patients with T2D and COPD.

Our study had a number of limitations. First, national health administrative data lacked complete information on family history, individual lifestyle, smoking, and alcohol consumption, which could have influenced the findings of our study. In any case, we had well-matched baseline characteristics (all SMD values <0.1) and tried to reduce the difference between the study and control groups. Second, this administrative database lacked results of hemoglobin A1C levels, biochemical tests, pulmonary function tests, and imaging methods, which precluded accurate evaluation of T2D and COPD severity. However, we utilized DCSI scores, antidiabetics, and cardiovascular medications as representative factors to assess T2D status and the clinical records of moderate and severe COPD exacerbations to assess COPD severity. Third, patients on insulin therapy often exhibit a long duration of T2D, suboptimal glycemic control, and more comorbidities. Although we considered these factors, unknown residual confounders could bias the results. Fourth, the participants in this study were mainly of Taiwanese descent; therefore, the results may not be applicable to other racial groups. Finally, in a retrospective cohort study, there are always unobserved confounders. Therefore, there is a need for randomized control trials to confirm our findings.

## 4. Materials and Methods

### 4.1. Data Source and Population of the Study

We identified patients from the Taiwan’s National Health Insurance Research Database (NHIRD), which was described in our previous study [25]. About 450 hospitals and over 10,000 clinics across Taiwan participate in the National Health Insurance program. The Health Insurance Authority conducts random inspections of all medical institutions annually to ensure the diagnostic accuracy and quality of care. The NHIRD has a link to the National Death Registry to confirm the diagnosis of death. The Declaration of Helsinki was followed for all methods used in this study. The China Medical University and Hospital Research Ethics Committee (CMUH104-REC2-115-CR4) approved this study. To prevent data leakage, detailed patient and provider information was encrypted and de-identified before release. The research ethics committee granted a waiver of informed consent.

### 4.2. Study Design

We identified persons with coexisting T2D and COPD from the NHIRD between 1 January 2000 and 31 December 2018, and followed them until 31 December 2019 (Figure 1). T2D and COPD were diagnosed with the ICD-9/10-CM codes for at least two outpatient visits or one admission in 1 year (Appendix A). Previous studies validated the ICD coding algorithms to define T2D and COPD with acceptable accuracy [37,38]. A moderate exacerbation of COPD was determined by the use of medication (systemic corticosteroids or antibiotics) and treatment in the outpatient claims; severe COPD exacerbations were defined as an acute attack treated at the emergency room (ER) or hospitalization [39]. Exclusion criteria were (1) age <18 or >80 years; (2) missing data of age or sex; (3) diagnosis of T1D, bacterial pneumonia, liver failure, or on dialysis; (4) hospitalization for COPD, bacterial pneumonia, ventilation use, or lung cancer, or death within 180 days of the index date to exclude latent morbidity; and (5) diagnosis of T2D or COPD prior to 1 January 2000 for the exclusion of prevalent disease.

### 4.3. Procedures

We defined the day of simultaneous diagnosis of T2D and COPD as the comorbid date (Figure 3). In our database, there are 49,837 people from the diagnosis of T2D to T2D and COPD, 51,596 people from the diagnosis of COPD to COPD and T2D. Patients who received insulin treatment after the comorbid date were defined as insulin users, whereas those who never received insulin treatment during the study period were defined as insulin non-users. The index date was defined as the first date of insulin use after the comorbid date. The index date of insulin non-user was considered to be the same period of time from the comorbid date to the index date of the insulin user. We assessed and matched the following variates that could interfere with the results: sex, age, obesity (the diagnoses of obesity, overweight, or severe obesity), smoking, comorbidities (hypertension, coronary artery disease, dyslipidemia, chronic kidney disease, atrial fibrillation, liver cirrhosis, and peripheral artery disease diagnosed one year before the index date), and medications (respiratory drugs, oral antidiabetics, antihypertensives, statins, and aspirin). Charlson Comorbidity Index (CCI) was assessed as a proxy for disease burden [40]. The Diabetes Complication Severity Index (DCSI) score [41] was used to assess complications of T2D; the occurrence of moderate or severe COPD exacerbations was used to assess COPD severity.

### 4.4. The Outcomes of Interest

The outcomes of interest in this study were all-cause mortality (the diagnosis of mortality was confirmed with the linkage to the National Death Registry), hospitalization for COPD, bacterial pneumonia, invasive mechanical ventilation (IMV), noninvasive positive pressure ventilation (NIPPV), lung cancers, and severe hypoglycemia (hypoglycemia requiring an emergency room visit or hospitalization). To calculate the outcomes, we followed patients until the date of death, respective outcomes, or at the end of follow-up on 31 December 2019, whichever came first.

### 4.5. Statistical Analyses

We used the Kelsey method and adopted the two-sided significance level of 95%, power of 80%, ratio of sample size between unexposed/exposed of 1, percentage of unexposed with outcome of 5%, and odds ratio of 1.5 to estimate the required exposed and unexposed sample size of 1690 and 1690, respectively [42]. Propensity score matching has been used for adjustment and comparability between insulin users and non-users [43]. We selected 32 clinically related variables as independent variates (including age, gender, comorbidities, DCSI and CCI scores, COPD exacerbation, and medications; Table 1) and insulin use as a dependent variable to estimate each patient’s propensity score using non-parsimonious multivariable logistic regression. The nearest neighbor algorithm has been used for the construction of matched pairs, and a standardized mean difference (SMD) <0.1 was considered a negligible difference between the study group and the control groups. Using the propensity score matching method, a total of 2370 pairs of insulin users and non-users were identified.

Chi-squared tests were used to determine statistical differences in categorical variables, and Student’s *t*-tests were used to determine statistical differences in continuous variables and between the study and control groups. To examine outcomes between insulin users and non-users, Cox proportional hazards models with multivariable adjustment were used. Results are presented as hazard ratios (HRs) and 95% confidence intervals (CIs). The Kaplan–Meier method and the log-rank test were used to compare the cumulative incidence of hospitalization for COPD, bacterial pneumonia, and invasive mechanical ventilation between insulin use and non-use. We have also compared the cumulative duration (<90, 90–179, >179 days) of insulin use and the risks of hospitalization for COPD, bacterial pneumonia, and invasive mechanical ventilation with insulin non-use. The cumulative duration of insulin was calculated by adding up from the index day to the end day of insulin use during the follow-up time.

Statistical significance was defined as a two-sided *p*-value of less than 0.05. Analyses of this study were performed using SAS (version 9.4; SAS Institute, Cary, NC, USA).

## 5. Conclusions

Patients with coexisting T2D and COPD require integrated, patient-centered, and multidisciplinary care. Pitifully, research on treatment strategies is scarce. Insulin may be indicated in diabetic patients with acute exacerbation of COPD to lower blood glucose. Our study has shown that patients with coexisting T2D and COPD requiring insulin therapy was associated with a higher risk of hypoglycemia, hospitalization for COPD, bacterial pneumonia, and ventilator use. However, we should be aware that this could be the result of the patients’ intrinsic severe metabolic decompensation, which led to the previous administration of insulin. Insulin administration requires effective doctor–patient communication for careful monitoring of hypoglycemia, pneumonia, acute exacerbation of COPD, and respiratory failure with immediate and appropriate disposal.

## Figures and Tables

**Figure 1 pharmaceuticals-16-00643-f001:**
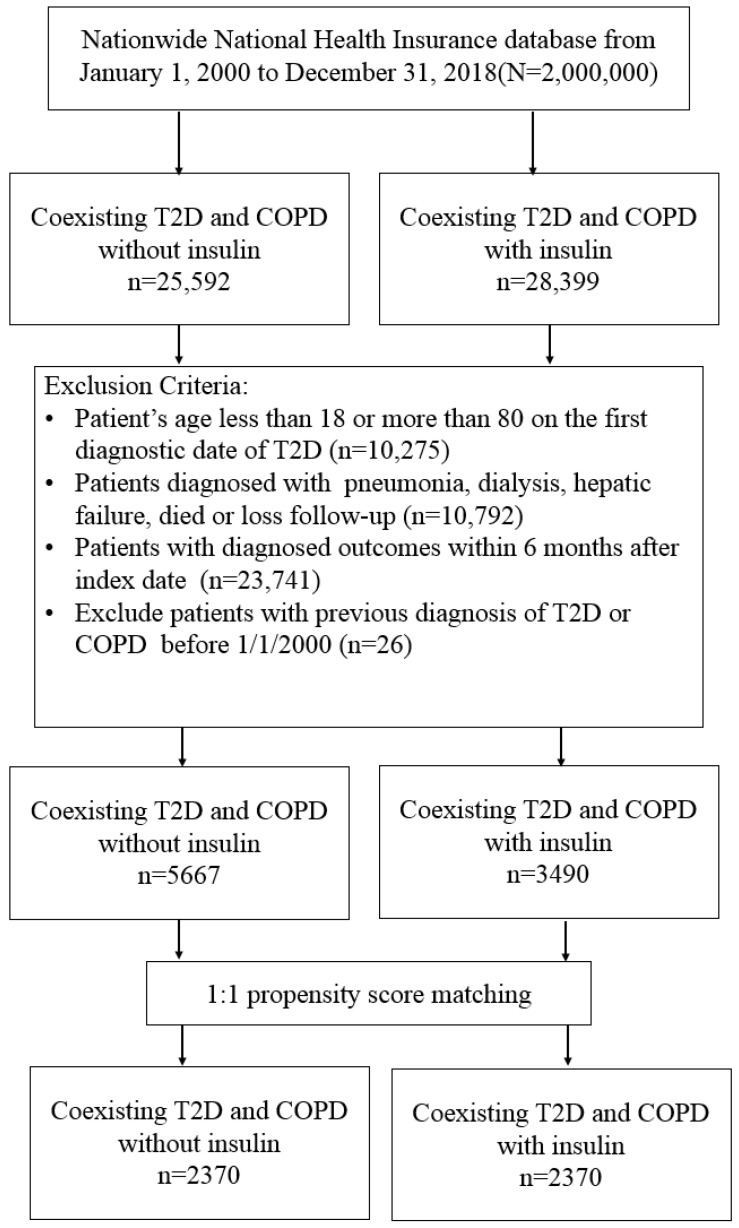
Patient selection flowchart.

**Figure 2 pharmaceuticals-16-00643-f002:**
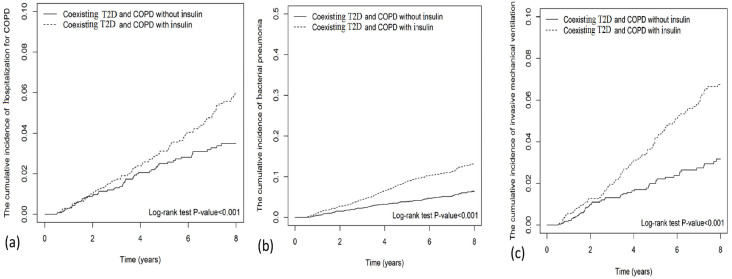
Cumulative incidence of (**a**) hospitalization for COPD, (**b**) bacterial pneumonia, and (**c**) invasive mechanical ventilation (IMV) between insulin use and no-use.

**Figure 3 pharmaceuticals-16-00643-f003:**
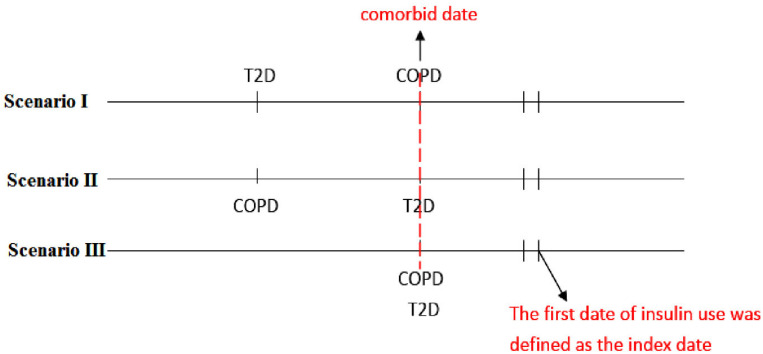
The scenarios of comorbid type 2 diabetes and COPD.

**Table 1 pharmaceuticals-16-00643-t001:** Comparison of variables in patients with coexisting T2D and COPD with and without insulin use.

	Non-Insulin		Insulin		
	(N = 2370)		(N = 2370)		
Variables	*n*	%	*n*	%	SMD ^§^
Sex					0.003
female	894	37.72	890	37.55	
male	1476	62.28	1480	62.45	
Age					
18–40	117	4.94	103	4.35	0.028
41–60	981	41.39	988	41.69	0.006
61–80	1272	53.67	1279	53.97	0.006
mean, (SD) ^†^	60.73	11.21	61.02	11.11	0.026
Comorbidities					
Obesity	60	2.53	63	2.66	0.008
Smoking	93	3.92	90	3.80	0.007
Hypertension	1541	65.02	1575	66.46	0.03
Dyslipidemia	1572	66.33	1610	67.93	0.034
Coronary artery disease	690	29.11	713	30.08	0.021
Atrial fibrillation	294	12.41	290	12.24	0.005
Peripheral arterial disease	94	3.97	96	4.05	0.004
Chronic kidney disease	323	13.63	342	14.43	0.023
Liver cirrhosis	47	1.98	47	1.98	<0.001
Moderate exacerbation of COPD	915	38.61	953	40.21	0.033
Severe exacerbation of COPD	1124	47.43	1153	48.65	0.024
Charlson Comorbidity Index					
1	2134	90.04	2134	90.04	<0.001
2–3	210	8.86	207	8.73	0.004
>3	26	1.10	29	1.22	0.012
Diabetes Complication Severity Index					
0	769	32.45	750	31.65	0.017
1	554	23.38	579	24.43	0.025
≥2	1047	44.18	1041	43.92	0.005
Medications					
Oral systemic corticosteroid	1723	72.70	1814	76.54	0.088
Corticosteroid inhalants	1954	82.45	2037	85.95	0.096
β2bronchodilators inhalants	29	1.22	34	1.43	0.018
Anticholinergic inhalants	150	6.33	145	6.12	0.009
Metformin	1332	56.20	1385	58.44	0.045
Sulfonylureas	1206	50.89	1247	52.62	0.035
Thiazolidinediones	311	13.12	315	13.29	0.005
Dipeptidyl peptidase-4 inhibitors	317	13.38	331	13.97	0.017
Sodium-glucose cotransporter 2 inhibitors	13	0.55	19	0.80	0.031
Alpha-glucosidase inhibitors	319	13.46	327	13.80	0.01
ACEI/ARB	1256	53.00	1284	54.18	0.024
β-blockers	735	31.01	722	30.46	0.012
Calcium-channel blockers	1353	57.09	1394	58.82	0.035
Diuretics	874	36.88	899	37.93	0.022
Statin	1032	43.54	1059	44.68	0.023
Aspirin	1055	44.51	1079	45.53	0.02

T2D, type 2 diabetes; COPD, chronic obstructive pulmonary disease; SD, standard deviation; SMD, standardized mean difference; ARB: angiotensin receptor blockers; ACEI: angiotensin-converting enzyme inhibitors. ^†^: Student’s *t*-test. ^§^: A negligible difference between the insulin and non-insulin cohorts is indicated by a SMD <0.1.

**Table 2 pharmaceuticals-16-00643-t002:** Hazard ratio (HR) and 95% confidence interval (CI) for the disease outcome in patients with T2D and COPD.

	Non-Insulin		Insulin									
Outcome	*n*	PY	IR	*n*	PY	IR	cHR	(95% CI)	*p*-Value	aHR ^a^	(95% CI)	*p*-Value
Death	326	15,198	21.45	357	16,026	22.28	1.04	(0.89, 1.2)	0.6482	1.08	(0.93, 1.26)	0.3076
Hospitalization for COPD	63	14,978	4.21	112	15,638	7.16	1.69	(1.24, 2.3)	<0.001	1.7	(1.24, 2.32)	<0.001
NIPPV	13	15,194	0.86	67	15,877	4.22	4.84	(2.67, 8.77)	<0.001	5.05	(2.76, 9.22)	<0.001
IMV	55	15,175	3.62	148	15,823	9.35	2.58	(1.89, 3.51)	<0.001	2.72	(1.99, 3.72)	<0.001
Bacterial pneumonia	122	14,855	8.21	286	15,154	18.87	2.29	(1.85, 2.83)	<0.001	2.42	(1.95, 3)	<0.001
Lung cancer	10	15,181	0.66	22	16,013	1.37	2.08	(0.99, 4.4)	0.0544	2.11	(0.99, 4.49)	0.0537
Hospitalized hypoglycemia	12	15,155	0.79	55	15,801	3.48	4.41	(2.36, 8.23)	<0.001	4.71	(2.5, 8.89)	<0.001

T2D, type 2 diabetes; COPD: Chronic obstructive pulmonary disease; PY: person-years; IR: incidence rate, per 1000 person-years; cHR, crude hazard ratio; aHR: adjusted hazard ratio; IMV: invasive mechanical ventilation NIPPV: Noninvasive positive pressure ventilation. aHR ^a^: multivariable analysis, including age, sex, comorbidities, and medications, as shown in Table 1.

**Table 3 pharmaceuticals-16-00643-t003:** The risk of outcomes in patients with a cumulative duration of insulin use or with no insulin use.

	Hospitalization for COPD						
Variables	*n*	PY	IR	cHR	(95% CI)	aHR ^a^	(95% CI)
Non-use of Insulin	63	14,978	4.21	1.00	(Reference)	1.00	(Reference)
Cumulative duration of insulin use (days)							
<90	67	10,998	6.09	1.44	(1.02, 2.04) *	1.49	(1.05, 2.12) *
90–179	14	1263	11.09	2.61	(1.46, 4.66) **	2.03	(1.12, 3.67) *
>179	31	3377	9.18	2.14	(1.39, 3.29) ***	2.15	(1.38, 3.35) ***
	Bacterial pneumonia						
Variables	*n*	PY	IR	cHR	(95% CI)	aHR	(95% CI)
Non-use of Insulin	122	14,855	8.21	1.00	(Reference)	1.00	(Reference)
Cumulative duration of insulin use (days)							
<90	155	10,778	14.38	1.75	(1.38, 2.22) ***	1.93	(1.52, 2.46) ***
90–179	42	1170	35.91	4.34	(3.06, 6.17) ***	3.35	(2.34, 4.8) ***
>179	89	3206	27.76	3.35	(2.55, 4.4) ***	3.45	(2.6, 4.58) ***
	Invasive mechanical ventilation						
Variables	*n*	PY	IR	cHR	(95% CI)	aHR	(95% CI)
Non-use of Insulin	55	15,175	3.62	1.00	(Reference)	1.00	(Reference)
Cumulative duration of insulin use (days)							
<90	76	11,134	6.83	1.88	(1.33, 2.67) ***	2.08	(1.46, 2.96) ***
90–179	20	1289	15.52	4.25	(2.55, 7.09) ***	3.31	(1.97, 5.59) ***
>179	52	3401	15.29	4.2	(2.87, 6.13) ***	4.38	(2.96, 6.47) ***

PY: person-years; IR: incidence rate, per 1000 person-years; cHR, crude hazard ratio; aHR: adjusted hazard ratio; COPD: Chronic obstructive pulmonary disease. aHR ^a^: multivariable analysis, including age, sex, comorbidities, and medications, as shown in Table 1. * *p* < 0.05, ** *p* < 0.01, *** *p* < 0.001.

## Data Availability

Data of this study are available from the National Health Insurance Research Database (NHIRD) published by Taiwan National Health Insurance (NHI) Administration. The data utilized in this study cannot be made available in the paper, the Appendix A, or in a public repository due to the “Personal Information Protection Act” executed by Taiwan government starting from 2012. Requests for data can be sent as a formal proposal to the NHIRD Office (https://dep.mohw.gov.tw/DOS/cp-2516-3591-113.html, accessed on 24 October 2022) or by email to stsung@mohw.gov.tw.

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
