# Peer review of "Respiratory Outcomes of Insulin Use in Patients with COPD: A Nationwide Population-Based Cohort Study"

_pharmaceuticals, 2023, doi:10.3390/ph16050643_

Round 1

Reviewer 1 Report (New Reviewer)

The manuscript entitled „ Respiratory Outcomes of Insulin Use in Patients with COPD: A Nationwide Population-Based Cohort Study” presents interesting issue, but some problems should be corrected.

It seems to be the corrected version of some previous manuscript, as it is prepared in changes tracking option. I can notice that some Authors of the previous version are now removed

Major:

1.       It is hard to understand what was in fact the objective of the study, as in various sections different ones are presented – in Abstract Authors stated that “We performed this study to examine the respiratory outcomes of insulin use versus no-use in persons with COPD.” (suggesting comparison of diabetic COPD patients using insulin with combined non-diabetic COPD ones and diabetic COPD ones but with no insulin), while in the Design section that “We identified persons with coexisting T2D and COPD”. Authors should be consistent within the whole study, as it is only one example, while in various sentences and areas they are also not consistent.

2.       The other problem results from the fact that Authors treat using insulin as an independent factor influencing the broaden consequences, but they must remember that there was a reason of using insulin. And this health-related reason may also have its other consequences. So, Authors must present more balanced opinion and can not indicate that using insulin is harmful, as those who used it within the study must have done it (without it they would have even more serious health-related consequences). Taking this into account, the sentences such as “This nationwide cohort study demonstrated that insulin use might increase the risks of acute COPD exacerbation, pneumonia, ventilation use, and severe hypoglycemia” should not be formulated.

Abstract:

Instead of what was one only, Authors should clearly identify their aim (e.g. “The aim of the study was…”).

The applied methodology should be clearly presented.

Conclusions should be properly formulated – see above

Introduction:

Instead of what was one only, Authors should clearly identify their aim (e.g. “The aim of the study was…”).

Materials and Methods:

Existing problems should be corrected – see above

The applied methodology should be clearly presented.

Results:

All the figures are of very low resolution and they are almost impossible to follow

The presentation of results should be proper – see above

Discussion:

The discussion of results should be proper – see above

Conclusions:

Conclusions should be properly formulated – see above

No references should be applied within conclusions

Author Response

Thank you for reviewing our manuscript and giving us insightful recommedations. We have tried to do our best to revise the manuscript as you suggested. 

Comments and Suggestions for Authors

The manuscript entitled „ Respiratory Outcomes of Insulin Use in Patients with COPD: A Nationwide Population-Based Cohort Study” presents interesting issue, but some problems should be corrected.

It seems to be the corrected version of some previous manuscript, as it is prepared in changes tracking option. I can notice that some Authors of the previous version are now removed

Major:

  1. It is hard to understand what was in fact the objective of the study, as in various sections different ones are presented – in Abstract Authors stated that “We performed this study to examine the respiratory outcomes of insulin use versus no-use in persons with COPD.” (suggesting comparison of diabetic COPD patients using insulin with combined non-diabetic COPD ones and diabetic COPD ones but with no insulin), while in the Design section that “We identified persons with coexisting T2D and COPD”. Authors should be consistent within the whole study, as it is only one example, while in various sentences and areas they are also not consistent.

Response: Thank you for reviewing our manuscript and giving us insightful recommendations. All participating patients had coexisting T2D and COPD, we only compared their respiratory outcomes between those with and without insulin treatment. We have reviewed the full text and made corrections to reduce confusion.

  1. The other problem results from the fact that Authors treat using insulin as an independent factor influencing the broaden consequences, but they must remember that there was a reason of using insulin. And this health-related reason may also have its other consequences. So, Authors must present more balanced opinion and can not indicate that using insulin is harmful, as those who used it within the study must have done it (without it they would have even more serious health-related consequences). Taking this into account, the sentences such as “This nationwide cohort study demonstrated that insulin use might increase the risks of acute COPD exacerbation, pneumonia, ventilation use, and severe hypoglycemia” should not be formulated.

Response: We agree with your opinion and try to present more balanced statement on page 1, 2, 9, and 11. 

Abstract:

Instead of what was one only, Authors should clearly identify their aim (e.g. “The aim of the study was…”).

The applied methodology should be clearly presented.

Conclusions should be properly formulated – see above

Response: We have modified the aim, the applied methodology and conclusions of Abstract on page 1 as”This nationwide cohort study showed that patients with T2D and COPD requiring insulin therapy may have an increased risk of acute COPD exacerbations, pneumonia, ventilator use, and severe hypoglycemia without a significant increase in the risk of death.” .

Introduction:

Instead of what was one only, Authors should clearly identify their aim (e.g. “The aim of the study was…”).

Response: We have modified the aim on page 2 as” Therefore, we conducted this cohort study to compare the outcomes of all-cause mortality, hospitalization for COPD, bacterial pneumonia, ventilator use, lung cancer, and severe hypoglycemia in people with coexisting T2D and COPD with or without receiving insulin therapy.”

Materials and Methods:

Existing problems should be corrected – see above

The applied methodology should be clearly presented.

Response: We have improved the statement of methodology on page 4 as suggested.  

Results:

All the figures are of very low resolution and they are almost impossible to follow

The presentation of results should be proper – see above

Response: We have upgraded the figures to high resolution (dpi more than 300) and attached the figures to the submission port.

Discussion:

The discussion of results should be proper – see above

Response: We have improved the discussion of results one page 8-10. 

Conclusions:

Conclusions should be properly formulated – see above

No references should be applied within conclusions

Response: Thanks for your recommendations! We have moved the cited paragraph, and changed the conclusions on page 11 as” Our study has shown that patients with coexisting T2D and COPD requiring insulin therapy was associated with a higher risk of hypoglycemia, hospitalization for COPD, bacterial pneumonia, and ventilator use. “

Reviewer 2 Report (New Reviewer)

Comments to Authors 

           This study showed that insulin use was associated with a higher risk of hypoglycemia, hospitalization for COPD, bacterial  pneumonia, and ventilator use.

          Type 2 diabetes mellitus (T2DM) and chronic obstructive pulmonary disease (COPD) often co-exist, yielding increased risk of cardiovascular (CV) complications including heart failure (HF). In patients with T2DM and CVD, COPD increased the risk of mortality and cardiorenal outcomes, including HF [1]. Empagliflozin consistently reduced these outcomes versus placebo regardless of COPD, suggesting that empagliflozin's benefits in patients with T2DM and CVD are not mitigated by the presence of COPD [1].

           Insulin-like growth factor-1 (IGF-1) display a vital role in in the pathogenesis of lung diseases, however, the relationship between circulating IGF-1 and lung disease remains unclear [2].

          Authors are kindly requested to emphasize the current concepts about these issues in the context of recent knowledge and the available literature. This articles should be quoted in the References list.

References

1.      Empagliflozin in patients with type 2 diabetes mellitus and chronic obstructive pulmonary disease. Diabetes Res Clin Pract. 2022; 186: 109837. doi:10.1016/j.diabres.2022.109837.

2.      Circulating insulin-like growth factor-1 and risk of lung diseases: A Mendelian randomization analysis. Front Endocrinol (Lausanne). 2023; 14: 1126397. Published 2023 Mar 3. doi:10.3389/fendo.2023.1126397.

Author Response

Thank you for your kind encouragement and nice recommendations. We have uploaded the response :

Comments to Authors

  1. This study showed that insulin use was associated with a higher risk of hypoglycemia, hospitalization for COPD, bacterial pneumonia, and ventilator use.

Response: Thank you for reviewing and encouraging us to keep going.

  1. Type 2 diabetes mellitus (T2DM) and chronic obstructive pulmonary disease (COPD) often co-exist, yielding increased risk of cardiovascular (CV) complications including heart failure (HF). In patients with T2DM and CVD, COPD increased the risk of mortality and cardiorenal outcomes, including HF [1]. Empagliflozin consistently reduced these outcomes versus placebo regardless of COPD, suggesting that empagliflozin's benefits in patients with T2DM and CVD are not mitigated by the presence of COPD [1].

Insulin-like growth factor-1 (IGF-1) display a vital role in in the pathogenesis of lung diseases, however, the relationship between circulating IGF-1 and lung disease remains unclear [2].

          Authors are kindly requested to emphasize the current concepts about these issues in the context of recent knowledge and the available literature. This articles should be quoted in the References list.

References

  1. Empagliflozin in patients with type 2 diabetes mellitus and chronic obstructive pulmonary disease. Diabetes Res Clin Pract. 2022; 186: 109837. doi:10.1016/j.diabres.2022.109837.

  1. Circulating insulin-like growth factor-1 and risk of lung diseases: A Mendelian randomization analysis. Front Endocrinol (Lausanne). 2023; 14: 1126397. Published 2023 Mar 3. doi:10.3389/fendo.2023.1126397.

Response: Thank you for your kind suggestion. We have highlighted these points on page 10-11, and cited the references.

Round 2

Reviewer 1 Report (New Reviewer)

Authors corrected their manuscript accordingly

Author Response

Thank you for your warm encouragement.

This manuscript is a resubmission of an earlier submission. The following is a list of the peer review reports and author responses from that submission.

Round 1

Reviewer 1 Report

Thank you for giving me the opportunity to read and comment a report “Respiratory Outcomes of Insulin Use in Patients With COPD: A Nationwide Population-based Cohort Study”, by Yen FS, et al.

In the reviewed manuscript, the respiratory outcomes of patients with COPD and with or without insulin use, has been evaluated.

This paper is well written, correctly structured with a suitable research concept, the study limitations are addressed, and it is of relevance to readers of the journal. However, I include a few comments for your consideration.

·         In the introduction section, the authors state the following: Insulin was discovered in 1921 and has been used to treat and save many children with type 1 diabetes, with little or no insulin production.

This statement is generic and not related to the study aim. It was also very useful for adult patients.

·         A more detailed description of the insulin use in COPD should be included in the Introduction section.

·         It would be desirable for the authors to provide more detail on the main aim of the study.

·  In Table 1, the authors mention having used Student's t-test, however this method is not described in the subsection "Statistical Analysis". Please describe it.

·   The graphs in Figure 3 are very small, making them difficult to understand. Please modify Figure 3 in this sense.

Author Response

Thank you for reviewing our manuscript and providing your insightful recommendations. We have attached the responses to your comments.

Thank you for giving me the opportunity to read and comment a report “Respiratory Outcomes of Insulin Use in Patients With COPD: A Nationwide Population-based Cohort Study”, by Yen FS, et al.

In the reviewed manuscript, the respiratory outcomes of patients with COPD and with or without insulin use, has been evaluated.

This paper is well written, correctly structured with a suitable research concept, the study limitations are addressed, and it is of relevance to readers of the journal. However, I include a few comments for your consideration.

Response: Thanks for reviewing our manuscript and giving us insightful recommendations.

  1. In the introduction section, the authors state the following: Insulin was discovered in 1921 and has been used to treat and save many children with type 1 diabetes, with little or no insulin production.

This statement is generic and not related to the study aim. It was also very useful for adult patients.

Response: We have modified this statement one page 2 as “Insulin was discovered in 1921 and has been used to treat and save many patients with type 1 diabetes, with little or no insulin production [11]. It also is the most effective glucose-lowering agent in adult persons with type 2 diabetes [12].”

  1. A more detailed description of the insulin use in COPD should be included in the Introduction section.

Response: We have added a more detailed description of the insulin use in COPD in the Introduction section as” Currently, there are no guidelines for managing diabetes mellitus in patients with coexisting T2D and COPD. The recommendation for antidiabetic treatment in these patients is in accordance with the guidelines for the management of diabetes mellitus [7,17]. Patients with acute exacerbations of COPD can be treated with oral antidiabetic drugs or glucagon-like peptide-1 agonists if their blood glucose levels are not very high. However, if they have severe hyperglycaemia, acute symptoms or require hospitalisation, they may need insulin or follow inpatient diabetes management guidelines [18]. Accordingly, they can be treated initially with long-acting insulin analogues because they are more stable, have a long-lasting effect, and carry a lower risk of glucose fluctuations and hypoglycaemia than neutral protamine Hagedorn insulin and premixed insulin [17]. Rapid-acting insulin analogues can then be added as needed, as they can rapidly lower blood glucose levels with a lower risk of postprandial hypoglycaemia than regular short-acting insulin [17]. Combined with tight titration of insulin doses and blood glucose monitoring, insulin therapy can be a safe and effective antidiabetic management strategy in patients with COPD and high blood glucose levels [7,18].”

  1. It would be desirable for the authors to provide more detail on the main aim of the study.

Response: We have modified this paragraph on page 2 as” We, therefore, conducted this cohort study aiming to compare the outcomes of mortality, hospitalisation for COPD, bacterial pneumonia, ventilator use, lung cancer and severe hypoglycaemia in patients with COPD with and without insulin use.”

  1. In Table 1, the authors mention having used Student's t-test, however this method is not described in the subsection "Statistical Analysis". Please describe it.

Response: We have described this method on page 7 as “The Student's t-test was used to determine statistical differences in continuous variables, and the chi-squared test was used to determine statistical differences in categorical variables between the study and control groups.”

  1. The graphs in Figure 3 are very small, making them difficult to understand. Please modify Figure 3 in this sense.

Response: Thanks! We have modified Figure 3 to make it more easy to understand.

Reviewer 2 Report

Authors conducted a very interesting study highlighting the importance of the link between DM and COPD. 

I have some comments:

- I don't see data about systemic corticosteroids usage between the two groups and their possible correlation. This is essential as it's well-known that CS can have a significant impact on blood sugar.  

-Moreover, data about inhaled corticosterioids and other inhaled therapy should be added and analyzed, too. 

- Was sample size estimated? Please specify it in data analysis section.

- I found several minor grammar errors throughout the manuscript. Please have a deep language revision.

Author Response

Thank you for reviewing our manuscript and providing your insightful recommendations. 

Authors conducted a very interesting study highlighting the importance of the link between DM and COPD.

I have some comments:

  1. I don't see data about systemic corticosteroids usage between the two groups and their possible correlation. This is essential as it's well-known that CS can have a significant impact on blood sugar. 

Response: Thanks for reviewing our manuscript and giving us meaningful recommendations. We have added the data of systemic corticosteroid use between the study and control groups and their possible correlation in Table 1.   

  1. Moreover, data about inhaled corticosteroids and other inhaled therapy should be added and analyzed, too.

Response: We have added the data of inhaled corticosteroid and other inhalers between the study and control groups and their possible correlation in Table 1. We have well-matched these variables between the two groups and adjusted these factors for the multivariate analysis of outcomes in Table 2 and Table 3.

  1. Was sample size estimated? Please specify it in data analysis section.

Response: We have specified the sample size estimation in the data analysis section on page 5 as “We used the Kelsey method and adopted the two-sided significance level (1-alpha) of 95%, power (1-beta, % chance of detection) of 80%, ratio of sample size between unexposed/exposed of 1, percentage of unexposed with outcome of 5%, and odds ratio of 1.5 to estimate the required exposed and unexposed sample size of 1690 and 1690, respectively [25].”  

  1. I found several minor grammar errors throughout the manuscript. Please have a deep language revision.

Response: We have performed a deep language revision of the whole manuscript to correct grammar errors. 

Reviewer 3 Report

It is a nice manuscript, with reasonable methodological neatness and well-established limitations and scope. However, It is suggested to clarify the following:

 In the abstract, the authors indicate: “Abstract: The acute exacerbation of chronic obstructive pulmonary disease (COPD) often requires insulin to lower blood glucose in patients with type 2 diabetes (T2D) and COPD. We conducted this cohort study to compare the respiratory outcomes of insulin use versus no use.

 1.- Was insulin administration to studied patients always associated with an exacerbation of COPD?

 Otherwise, it is suggested to describe the main causes of the previous administration of insulin. i.e. severe hyperglycemia (random glucose > 300 mg/dl), symptomatic diabetes, ketoacidosis, acute medical events, concomitant diseases, admission, and inability to take oral antidiabetic drugs or if simply the patient's glycemia was being routinely controlled with insulin.

 2.- Based on the above, it is suggested briefly discuss what would have happened if these patients were not given insulin. Was there any other alternative?

 3.- Notwithstanding the methodological rigor used, there could be doubts about insulin causality on the differences found between groups versus the impact of the intrinsic great metabolic decompensation of the patients that gave rise to the previous administration of insulin. It is suggested also to consider this last possibility in the conclusion section.

Author Response

Thank you for reviewing our manuscript and providing your insightful recommendations. 

It is a nice manuscript, with reasonable methodological neatness and well-established limitations and scope. However, It is suggested to clarify the following:

 In the abstract, the authors indicate: “Abstract: The acute exacerbation of chronic obstructive pulmonary disease (COPD) often requires insulin to lower blood glucose in patients with type 2 diabetes (T2D) and COPD. We conducted this cohort study to compare the respiratory outcomes of insulin use versus no use.

 1.- Was insulin administration to studied patients always associated with an exacerbation of COPD? Otherwise, it is suggested to describe the main causes of the previous administration of insulin. i.e. severe hyperglycemia (random glucose > 300 mg/dl), symptomatic diabetes, ketoacidosis, acute medical events, concomitant diseases, admission, and inability to take oral antidiabetic drugs or if simply the patient's glycemia was being routinely controlled with insulin.

Response: Thanks for your encouragement and recommendation. We agree with your opinion that patients with acute exacerbations of COPD are not always treated with insulin, but are treated according to DM guidelines for the management of these patients with hyperglycaemia. We have modified this portion on page 2 as” Currently, there are no guidelines for managing diabetes mellitus in patients with coexisting T2D and COPD. The recommendation for antidiabetic treatment in these patients is in accordance with the guidelines for the management of diabetes mellitus [7,17] “.

 2.- Based on the above, it is suggested briefly discuss what would have happened if these patients were not given insulin. Was there any other alternative?

Response: We have modified this paragraph one page 2 as” Patients with acute exacerbations of COPD can be treated with oral antidiabetic drugs or glucagon-like peptide-1 agonists if their blood glucose levels are not very high. However, if they have severe hyperglycaemia, acute symptoms or require hospitalisation, they may need insulin or follow inpatient diabetes management guidelines [18]”.

 3.- Notwithstanding the methodological rigor used, there could be doubts about insulin causality on the differences found between groups versus the impact of the intrinsic great metabolic decompensation of the patients that gave rise to the previous administration of insulin. It is suggested also to consider this last possibility in the conclusion section.

Response: Thanks for your recommendation. We have added the statement on page 10 as” However, we should be aware that this could be the result of the patients' intrinsic severe metabolic decompensation, which led to the previous administration of insulin.”

Reviewer 4 Report

As far as I know, the publication submitted for review is already after one or several rounds of revisions. Overall, the manuscript is very carefully prepared.

I have a few minor comments:

1. Figures should be of better quality.

2. The authors could cite the following manuscripts in the discussion:

a. The Impact of Diabetes Mellitus in Patients with Chronic Obstructive Pulmonary Disease (COPD) Hospitalization

b. Mechanisms Linking COPD to Type 1 and 2 Diabetes Mellitus: Is There a Relationship between Diabetes and COPD?

Author Response

Thank you for reviewing our manuscript and providing your insightful recommendations. 

As far as I know, the publication submitted for review is already after one or several rounds of revisions. Overall, the manuscript is very carefully prepared.

I have a few minor comments:

  1. Figures should be of better quality.

Response: Thank you for reviewing our manuscript and for your encouragement. We have improved the figures for better quality. 

  1. The authors could cite the following manuscripts in the discussion:

  1. The Impact of Diabetes Mellitus in Patients with Chronic Obstructive Pulmonary Disease (COPD) Hospitalization
  2. Mechanisms Linking COPD to Type 1 and 2 Diabetes Mellitus: Is There a Relationship between Diabetes and COPD?

Response: We have cited these references in the Introduction and Discussion sections.

Reviewer 5 Report

Reviewer comments and suggestions

The author conducted a cohort study to compare the respiratory outcomes of insulin use versus no use. Propensity score matching was used to identify 2370 pairs of insulin users and nonusers from Taiwan’s National Health Insurance. The mean follow-up time for insulin users and nonusers was 6.65 and 6.37 years, respectively. Compared with no-use of insulin, insulin use was associated with a significantly higher risk of hospitalization for COPD (aHR 1.65), and other conditions but without a significant difference in the risk of all-cause mortality. Finally, the study suggested that insulin may increase the risks of acute COPD exacerbation, pneumonia, and other condition. However, without significantly increasing the risk of death in patients with T2D and COPD.

Below are the comments for this paper to be incorporated in the revised version of the manuscript. 

  1. “ and low-grade systemic inflammation in patients with COPD also predisposes them to T2D [1,7,8]” Explore these studies
  2. is there any possible reason for this “diabetes that negatively influences pulmonary function and survival in patients with cystic fibrosis [14]”
  3. ”However, few studies have examined the long-term outcomes of insulin use in patients with COPD” need references here
  4. Figures 1 and 3 please paste the original figure
  5. did the authors check how many type 2 diabetic participants were diagnosed with COPD and vice verse
  6. need to explore it well, reason does not fit well “increase, insulin “toxicity” and insulin resistance, which could suppress autophagy, favor senescence, decrease pulmonary function, and exacerbate severe COPD exacerbation [8,24,25].”
  7. There were several places the author has written previous studies, several studies but cite only one reference. Either they change the sentence or cite more references, please check the whole MS
  8. Smoking is the main cause of lung cancer [1]. Both COPD and T2D are risk factors for lung cancer [1,8,22], The authors could find more references other than 1 reference. These types of errors should also minimize.

Author Response

Thank you for reviewing our manuscript and providing your insightful recommendations. 

The author conducted a cohort study to compare the respiratory outcomes of insulin use versus no use. Propensity score matching was used to identify 2370 pairs of insulin users and nonusers from Taiwan’s National Health Insurance. The mean follow-up time for insulin users and nonusers was 6.65 and 6.37 years, respectively. Compared with no-use of insulin, insulin use was associated with a significantly higher risk of hospitalization for COPD (aHR 1.65), and other conditions but without a significant difference in the risk of all-cause mortality. Finally, the study suggested that insulin may increase the risks of acute COPD exacerbation, pneumonia, and other condition. However, without significantly increasing the risk of death in patients with T2D and COPD.

Below are the comments for this paper to be incorporated in the revised version of the manuscript.

1.“and low-grade systemic inflammation in patients with COPD also predisposes them to T2D [1,7,8]” Explore these studies.

Response: Thank you for reviewing our manuscript and providing insightful recommendations.

We have explored this part on page 2 as” People with COPD have elevated levels of plasma CRP and activation of nuclear factor-κB (NF-κB), which can attenuate insulin suppression of hepatic glucose production and induce peripheral tissue insulin resistance. Pulmonary inflammation can spill over into systemic inflammation and adversely affect systemic glucose homeostasis by reducing the recruitment of skeletal muscle capillaries that deliver glucose and insulin to muscle cells, thereby increasing blood glucose levels [1,7-9].”

  1. Is there any possible reason for this “diabetes that negatively influences pulmonary function and survival in patients with cystic fibrosis [14]”

Response: We have explained this part on page 2 as” Patients with cystic fibrosis have less insulin secretion, are more likely to have diabetes and require insulin therapy. They often have poorer nutrition, greater catabolism and a higher risk of infection. This can have a negative impact on their lung function and survival [16].”

  1. ”However, few studies have examined the long-term outcomes of insulin use in patients with COPD” need references here

Response: We have modified this sentence on page 2 as” However, no studies have examined the long-term outcomes of insulin use in persons with COPD [7-9].”

  1. Figures 1 and 3 please paste the original figure

Response: We pasted the original Figure 1 and 3. 

  1. Did the authors check how many type 2 diabetic participants were diagnosed with COPD and vice versa.

Response: In our database, there are 49,837 people from the diagnosis of T2D to T2D and COPD, 51,596 people from the diagnosis of COPD to COPD and T2D. We have added this information on page 4.

  1. Need to explore it well, reason does not fit well “increase, insulin “toxicity” and insulin resistance, which could suppress autophagy, favor senescence, decrease pulmonary function, and exacerbate severe COPD exacerbation [8,24,25].”

Response: We have explored this part on page 9-10 as” Insulin treatment has a direct effect on the lungs by inducing prostaglandin-mediated contraction of airway smooth muscle [29]. Using insulin is likely to cause hypoglycaemia and weight gain [8,30]. Prolonged use of insulin can lead to hyperinsulinemia, downregulation of insulin signalling and insulin resistance, which can increase glucose levels and oxidative stress. The downregulated insulin signalling results in an unbalanced anabolic activity of insulin, favoring protein synthesis while suppressing autophagy. The latter inhibits the autophagic removal and turnover of proteins and lipids, which favors cell senescence [31]. All these factors can worsen lung function and exacerbate COPD.”

  1. There were several places the author has written previous studies, several studies but cite only one reference. Either they change the sentence or cite more references, please check the whole MS

Response: We have corrected these errors over the whole manuscript.

  1. Smoking is the main cause of lung cancer [1]. Both COPD and T2D are risk factors for lung cancer [1,8,22], The authors could find more references other than 1 reference. These types of errors should also minimize.

Response: We have added more references to this statement and minimized these errors in other areas.

Round 2

Reviewer 2 Report

Authors satisfactorily replied to my comments.

Therefore, IMHO this article can be now accepted.

Author Response

Thank you for reviewing our manuscript and giving us so much encouragement.